# A Machine Learning-Based Diagnostic Model for Crohn’s Disease and Ulcerative Colitis Utilizing Fecal Microbiome Analysis

**DOI:** 10.3390/microorganisms12010036

**Published:** 2023-12-24

**Authors:** Hyeonwoo Kim, Ji Eun Na, Sangsoo Kim, Tae-Oh Kim, Soo-Kyung Park, Chil-Woo Lee, Kyeong Ok Kim, Geom-Seog Seo, Min Suk Kim, Jae Myung Cha, Ja Seol Koo, Dong-Il Park

**Affiliations:** 1Department of Bioinformatics, Soongsil University, Seoul 06978, Republic of Korea; blissfulwooji@gmail.com (H.K.); sskimb@ssu.ac.kr (S.K.); 2Department of Internal Medicine, College of Medicine, Inje University Haeundae Paik Hospital, Busan 48108, Republic of Korea; h00609@paik.ac.kr (J.E.N.); kto0440@paik.ac.kr (T.-O.K.); 3Division of Gastroenterology, Department of Internal Medicine and Inflammatory Bowel Disease Center, Kangbuk Samsung Hospital, School of Medicine, Sungkyunkwan University, Seoul 03181, Republic of Korea; sk0103.park@samsung.com; 4Medical Research Institute, Kangbuk Samsung Hospital, School of Medicine, Sungkyunkwan University, Seoul 03181, Republic of Korea; chilwoo.lee@gmail.com; 5Department of Internal Medicine, College of Medicine, Yeungnam University, Daegu 42415, Republic of Korea; cello7727@naver.com; 6Department of Internal Medicine, School of Medicine, Wonkwang University, Iksan 54538, Republic of Korea; medsgs@wku.ac.kr; 7Department of Human Intelligence and Robot Engineering, Sangmyung University, Cheonan-si 31066, Republic of Korea; minsuk.kim@smu.ac.kr; 8Department of Internal Medicine, Kyung Hee University Hospital at Gangdong, Kyung Hee University College of Medicine, Seoul 05278, Republic of Korea; clicknox@khnmc.or.kr; 9Division of Gastroenterology and Hepatology, Department of Internal Medicine, Ansan Hospital, Korea University College of Medicine, Ansan 15355, Republic of Korea; jskoo@korea.ac.kr

**Keywords:** inflammatory bowel disease, Crohn’s disease, ulcerative colitis, fecal microbiome, sparse partial least squares discriminant analysis, machine learning

## Abstract

Recent research has demonstrated the potential of fecal microbiome analysis using machine learning (ML) in the diagnosis of inflammatory bowel disease (IBD), mainly Crohn’s disease (CD) and ulcerative colitis (UC). This study employed the sparse partial least squares discriminant analysis (sPLS-DA) ML technique to develop a robust prediction model for distinguishing among CD, UC, and healthy controls (HCs) based on fecal microbiome data. Using data from multicenter cohorts, we conducted 16S rRNA gene sequencing of fecal samples from patients with CD (*n* = 671) and UC (*n* = 114) while forming an HC cohort of 1462 individuals from the Kangbuk Samsung Hospital Healthcare Screening Center. A streamlined pipeline based on HmmUFOTU was used. After a series of filtering steps, 1517 phylotypes and 1846 samples were retained for subsequent analysis. After 100 rounds of downsampling with age, sex, and sample size matching, and division into training and test sets, we constructed two binary prediction models to distinguish between IBD and HC and CD and UC using the training set. The binary prediction models exhibited high accuracy and area under the curve (for differentiating IBD from HC (mean accuracy, 0.950; AUC, 0.992) and CD from UC (mean accuracy, 0.945; AUC, 0.988)), respectively, in the test set. This study underscores the diagnostic potential of an ML model based on sPLS-DA, utilizing fecal microbiome analysis, highlighting its ability to differentiate between IBD and HC and distinguish CD from UC.

## 1. Introduction

Ulcerative colitis (UC) and Crohn’s disease (CD), which constitute inflammatory bowel disease (IBD), are characterized by chronic inflammation of the intestines [1]. The current diagnostic approach for IBD involves a comprehensive strategy that involves medical history, blood and stool analyses, endoscopy with histological findings, and radiological imaging. However, these methods have inherent limitations as they rely on subjective interpretations without any gold standards and must rule out diseases that appear as IBD, leading to inconsistent results [2,3,4,5]. Consequently, the complexity of the diagnostic processes and the absence of specific markers often result in a median time to diagnosis of 3.7 months for UC and 8.0 months for CD, with diagnosis delays exceeding 6.7 and 15.2 months for UC and CD, respectively [6,7,8]. Unfortunately, the disease can progress rapidly and present acute exacerbation, leading to disease-related complications such as stricturing or penetrating disease, necessitating intestinal surgery [6,7,8]. Therefore, timely diagnosis is crucial for the initiation of effective treatment. 

Recently, interest in the role of the gut microbiome in IBD pathogenesis has increased [9,10,11]. Emerging evidence suggests that alterations in the composition and function of the gut microbiome contribute to the progression and therapeutic response of IBD [12]. This potential link between the gut microbiome and IBD underscores the need for innovative diagnostic tools that utilize fecal microbiome analysis as a noninvasive and easily accessible approach. These notions have been reinforced by numerous studies that have identified alterations in microbial diversity and specific bacterial taxa in patients with IBD compared with those in healthy individuals [13,14,15,16]. Distinctions between the fecal microbiomes of patients with UC and CD have been reported, suggesting the possibility of a classification based on these differences [13,14]. Furthermore, machine learning (ML) models have shown promising performance in distinguishing between patients with IBD and healthy individuals and between UC and CD [14,17,18,19]. These tools may help differentiate between individuals with IBD and those who are healthy and distinguish between UC and CD, two subtypes of IBD.

In contrast to ML algorithms utilized in previous studies, such as random forest (RF), sparse partial least squares discriminant analysis (sPLS-DA) has several advantages. The primary benefit of sPLS-DA is its ability to select a subset of informative variables to discriminate between classes. Additionally, choosing a sparse set of features helps manage many variables that may not contribute meaningfully to the classification task. Moreover, selecting variables with the most discriminative power can contribute to the creation of an interpretable model. 

No studies have used sPLS-DA to differentiate between patients with IBD and healthy controls (HCs) or between patients with UC and CD. Therefore, we implemented a prediction model using ML with sPLS-DA to distinguish between both IBD and HC and UC and CD, demonstrating its performance [20].

## 2. Materials and Methods

### 2.1. Research Cohorts and Sample Collection

We enrolled two patient cohorts, one comprising individuals with UC and the other comprising patients with CD, along with a cohort of healthy controls (HCs). The present study was undertaken in parallel with a retrospective multicenter study performed by an IMPACT (identification of the mechanism of CD occurrence and progression through an integrated analysis of both genetic and environmental factors) [21]. In 2017, the IMPACT study team was established in Korea and obtained a national grant to organize a retrospective cohort of patients with CD (aged > 8 years) to identify the mechanisms underlying the occurrence and progression of CD. A total of 16 university hospitals are currently participating in this study and collecting clinical data and biological specimens (namely blood, stool, and tissue specimens) from patients with CD who were newly diagnosed or followed up at these institutions. Patients with UC were selected from a prospective multicenter inception cohort study established for UC multi-omics research in Korea in 2020. Fourteen university hospitals participated in this study and collected clinical data and biological specimens, namely blood, stool, tissue, and saliva samples, from patients with UC. Lastly, the HC group consisted of healthy men and women aged 28–78 years who underwent regular health checkups, including body mass index, smoking status, alcohol consumption, and basic blood tests, annually or biennially at the Kangbuk Samsung Healthcare Screening Center from June to September 2014. This cohort comprised individuals who reported the absence of specific diseases using a self-report questionnaire. Further details are provided in a previous study [22]. An HC dataset was acquired by communicating with the authors.

Fecal samples were collected by participants (5 g each) and immediately stored in a deep freezer at −80 °C after submission. The collection time for the UC cohort as an inception cohort was the date of research registration before the initiation of drug therapy. Meanwhile, for the CD cohort with a retrospective design, wherein the patients were already diagnosed and were undergoing treatment, the collection times varied. To minimize these effects, fecal samples were collected after more than 3 months of discontinuing antibiotics or probiotics if the patient was taking them.

### 2.2. Sample Preparation and 16S rRNA Gene Sequencing

Information regarding sample preparation and sequencing can be found in a previous report [23]. Briefly, the samples were centrifuged at 15,000 rpm for 20 min at 4 °C to separate the cellular pellet from the cell-free supernatant. DNA was extracted from the cellular pellet using a QIAamp DNA Microbiome Kit (Qiagen, Valencia, CA, USA) in accordance with the manufacturer’s instructions.

For 16S rRNA amplicon sequencing, we targeted the high-resolution V3-V4 region, which is identical to the existing HC dataset [22] for comparability. The 16S rRNA gene’s V3-4 region was amplified with Illumina adapter overhang sequences using 341F (5′-TCG TCG GCA GCG TCA GAT GTG TAT AAG AGA CAG CCT ACG GGN GGC WGC AG-3′) and 805R (5′-GTC TCG TGG GCT CGG AGA TGT GTA TAA GAG ACA GGA CTA CHV GGG TAT CTA ATC C-3′) primers. PCR-generated amplicons were purified using a magnetic bead-based system (Agencourt AMPure XP; Beckman Coulter, Brea, CA, USA). Indexed libraries were prepared by limited-cycle PCR using the Nextera technology, cleaned, and pooled at equimolar concentrations. Paired-end sequencing was performed on an Illumina MiSeq platform using a 2 × 300 bp protocol, according to the manufacturer’s instructions.

### 2.3. Data Processing and Downstream Analysis

We employed a streamlined pipeline [24] based on HmmUFOtu (version 1.5.1) [25] to analyze the 16S rRNA amplicon sequencing data, as described below. Quality filtering of raw sequence data was performed using fastp [26]. Following the recommendations of fastp (version 0.23.2), sequences with a quality score below 20 and reads with a length of less than 150 bp were excluded, as described in a previous study [24] for HC sample processing using fastp. To perform reference-based operational taxonomic unit (OTU) clustering, each trimmed read was individually aligned to the HmmUFOtu model to generate a continuously aligned sequence for each pair. Subsequently, the contig sequences were positioned onto the reference phylogenetic tree (derived from GreenGene version 13.8 and the RDP Classifier Training set version 18) and assigned to the nearest node using the HmmUFOtu main program. The Biostrings (version 2.54.0) Bioconductor package was employed to generate consensus sequences by aggregating the amplicons associated with a shared HmmUFOtu node. We employed Mothur (version 1.48.0) [27] for de novo chimera checking of the consensus sequences, Kraken2 (version 2.1.2) [28] with default parameters, and SILVA reference (version 138.1) for taxonomic assignment.

Microbiome profile data were analyzed using phyloseq (version 1.38.0), a Bioconductor R package. Non-bacterial sequences and those lacking phylum-level annotations were excluded from the analysis. In subsequent analyses, we utilized the cut-off that had yielded significant results in an earlier study [24], excluding samples with fewer than 20,000 read counts and rarely observed phylotypes. We used the Bioconductor R package microbiome (version 1.16.0) to compute the alpha diversity indices for the samples. Using Mothur, we calculated beta diversity indices and conducted permutational multivariate analysis of variance (PERMANOVA) tests based on distance matrices to examine the differences in microbiome composition between different phenotypes.

### 2.4. Machine Learning for Disease Prediction Model

Given the merging of the datasets sequenced at different time points, we used ANCOM-BC (version 1.4.0) [29], specifying the covariate as the time point to adjust for batch effects among the sample groups sequenced at different times before constructing the ML model. We identified the fractions of taxonomic groups with significantly different absolute abundances at each time point. Subsequently, to mitigate variations owing to differences in sequencing depth among samples, we performed a log transformation by adding a pseudo-count of one and subtracting this fraction from the log-transformed abundance obtained from ANCOM-BC.

For subsequent steps, such as principal component analysis (PCA) and prediction model development, we used the mixOmics R package (version 6.18.1) in Bioconductor. We utilized sPLS-DA for variable selection, interpretable results, and computational efficiency.

Because our data were somewhat imbalanced, we matched the age, sex, and number distribution of each class group by downsampling the dataset before training the ML model. The dataset was then randomly divided into 70% training and 30% test sets while maintaining the class proportions.

We employed feature selection and parameter optimization, as recommended by mixOmics. First, we trained the initial sPLS-DA models and assessed their performance with 50 repeated 5-fold cross-validations (5-CVs) to determine the optimal number of components by monitoring the overall error rate trend. Subsequently, we performed tuning processes to select the features for each component. Using these optimal parameters, the final sPLS-DA model was developed, and its performance was measured.

To avoid bias or loss of information, the entire model development process, including downsampling, was repeated 100 times with random shuffling of the training and test splits. Subsequently, the average performance was assessed.

## 3. Results

### 3.1. Processing of 16S rRNA Gene Amplicon Sequencing Data

We performed 16S rRNA gene amplicon sequencing of stool samples from 2247 individuals, constituting three phenotypic groups: 671 with CD, 114 with UC, and 1462 HCs. The characteristics of each group are presented in Table 1.

During sequencing, we obtained 164,539,577 paired-end reads. After quality control, 157,961,202 reads remained. Following reference-based OTU clustering, we identified 88,927 OTUs. Taxonomic assignment and phylotyping of the remaining 83,562 OTUs after chimera removal led to the identification of 2525 phylotypes. In the abundance table filtering step, we filtered out phylotypes with abundances less than 10, those that did not belong to bacterial taxa, or those lacking specific phylum information from the entire dataset. Additionally, samples with a total abundance of less than 20,000 were excluded, resulting in 1517 phylotypes and 1846 samples. We used this dataset (CD, *n* = 670; UC, *n* = 113; HC, *n* = 1063) for subsequent analyses. Detailed information regarding each processing step is presented in Table 2.

### 3.2. Diversity Analysis

The results of the alpha diversity analysis showed that the stool microbiome in HC individuals was significantly richer than that in CD (*p* < 1 × 10^−2^) and UC (*p* < 1 × 10−^4^) patients (Figure 1a,b); however, between CD and UC, the alpha diversity indices were not significantly different.

Beta diversity principal coordinate analysis (PCoA) plots based on Jaccard and thetaYC dissimilarity indices (Figure 1c,d) showed a distinct separation between the IBD and HC samples along the PCoA1 axis, although there were some overlaps. In contrast, the CD and UC samples remained indistinguishable based on components 1 and 2 in both plots.

### 3.3. Multiclass Disease Prediction Model

Before model development, we conducted a log transformation and bias correction of the stool microbiome profile data using ANCOM-BC. To correct for the bias introduced by different sequencing time points in the profile data, we specified the input covariate of ANCOM-BC as the time-point information (seven time points). Taxonomic groups with significantly different absolute abundances at each time point were identified. Subsequently, we added a pseudo-count of one to the profile data, performed a log transformation, and subtracted the fraction obtained from the ANCOM-BC results.

Initially, we employed the sPLS-DA algorithm to create a multiclass ML model. The entire dataset (CD: *n* = 670, UC: *n* = 113, HC: *n* = 1063) was downsampled to match the age, sex distribution, and class counts (CD: *n* = 113, UC: *n* = 113, HC: *n* = 113) and then split into training and test sets with equal class balance. We allocated 70% of the samples to the training set (CD, *n* = 79; UC, *n* = 79; HC, *n* = 79), and 30% were assigned to the test set (CD, *n* = 34; UC, *n* = 34; HC, *n* = 34). This process was repeated 100 times to demonstrate the robustness of the model. In each repetition, the sPLS-DA model of the training set was initialized to identify the optimal components by monitoring the overall error rate. Subsequently, the tuning process selected the best features for each component. We defined the final sPLS-DA model for each run using these optimal components and phylotypes, and we evaluated the performance of each model using the corresponding test set.

Overall, these multiclass models showed suboptimal performances in classifying CD and UC, although the HC group was distinctly identified (Table 3 and Figure 2).

### 3.4. Hierarchical Disease Prediction Model

We chose to create two binary prediction models by observing the suboptimal performance of the multiclass model. The first distinguished IBD from HC samples, and the second classified IBD samples as CD or UC. This hierarchical approach enabled accurate classification of the three phenotypes.

#### 3.4.1. Creating a Predictive Model for Distinguishing between IBD and HC

The entire dataset (CD, *n* = 670; UC, *n* = 113; HC, *n* = 1063) was transformed into a binary classification dataset to distinguish between IBD and HC samples. Initially, 113 CD and 226 HC samples were selected and matched for age and sex with the UC samples. Subsequently, the dataset was divided to yield a 70% training set (CD, *n* = 79; UC, *n* = 79; HC, *n* = 158) and a 30% test set (CD, *n* = 34; UC, *n* = 34; HC, *n* = 68). The CD and UC samples in both sets were merged into the IBD class to form training (IBD, *n* = 158; HC, *n* = 158) and test sets (IBD, *n* = 68; HC, *n* = 68). This process was iterated 100 times using the same ML procedure applied to each split. The model performance was subsequently averaged across splits to provide a comprehensive evaluation.

As shown in Figure 3a, a representative final model produced a plot with a clear distinction between IBD and HC samples. The performance of each model was evaluated by predicting the disease class of individuals in the corresponding test sets. The IBD versus HC prediction achieved a mean accuracy of 0.950 (0.890–0.993), sensitivity of 0.918 (0.809–0.985), specificity of 0.985 (0.918–1), and precision of 0.984 (0.910–1) (Table 4).

We assessed the abundance of the top 10 phylotypes (Appendix A) that played key roles in predicting both the IBD and HC groups in the test set using a heatmap. Except for a few samples, we noticed that there was distinct clustering based on class using the 10 phylotype abundance criteria (Figure 3b).

#### 3.4.2. Creating a Predictive Model for Distinguishing between CD and UC

In the original 100 splits mentioned in Section 3.4.1, we exclusively selected CD and UC samples to establish training sets (CD, *n* = 79; UC, *n* = 79) and test sets (CD, *n* = 34; UC, *n* = 34) to develop models aimed at distinguishing between CD and UC. These datasets were utilized for model development and evaluation, following an earlier procedure.

A representative split sample plot showed a clear separation between the CD and UC samples (Figure 4a), indicating effective differentiation using stool microbiome data. We conducted a performance evaluation of the trained sPLS-DA models by predicting the disease phenotypes of individuals in the test sets. Across the 100 test sets, the classification results displayed a mean accuracy of 0.956, sensitivity of 0.941, specificity of 0.949, precision of 0.950, and AUC of 0.923 (Table 5). These results indicated that the fecal microbiome-based model could distinguish between CD and UC with excellent performance.

Using a heat map, we examined the abundance of the top 10 phylotypes (Appendix A) that were crucial for predicting both the CD and UC groups in the test set. We observed a distinct clustering based on class using the abundance criteria for the ten phylotypes, except for a few samples (Figure 4b).

#### 3.4.3. Performance Evaluation of Models in Hierarchical Manner

In the previous step, we noted the effectiveness of the fecal microbiome-based binary classification model in distinguishing IBD from HC, CD, and UC. We evaluated the performance of a hierarchical approach that integrates the two models to predict unknown class labels in the input samples. First, the samples were classified as either IBD or HC; then, among those categorized as IBD, further classification into CD or UC was performed. This hierarchical model was evaluated using test sets to assess its effectiveness. Table 6 presents the results obtained from 100 test sets, showing a mean accuracy of 0.936. It also reveals specific values for CD sensitivity of 0.888, CD precision of 0.965, UC sensitivity of 0.933, UC precision of 0.964, HC sensitivity of 0.956, and HC precision of 0.891.

## 4. Discussion

This study demonstrated the effectiveness of an ML model based on sPLS-DA, utilizing fecal microbiome data, in distinguishing between individuals with IBD and HC, as well as in differentiating between CD and UC. First, we constructed a multiclass ML model to differentiate among CD, UC, and HC. It performed well in distinguishing HC from IBD (CD or UC) with a mean sensitivity and precision of 0.952 and 0.814, respectively. However, it performed poorly in differentiating between CD and UC, with a sensitivity and precision <0.5. To overcome this limitation, we restructured two binary classification models in the next step: one to distinguish IBD from HC and the other to distinguish CD from UC. Using binary classification models, the AUC for distinguishing IBD from HC and CD from UC were outstanding, with values of 0.992 and 0.988, respectively. These findings have substantial implications as they demonstrate robust predictive capabilities.

The strength of this study lies in the pioneering use of sPLS-DA to construct a prediction model for distinguishing between IBD and HC, as well as between UC and CD. The sPLS-DA method employed in this study offers several advantages over conventional ML approaches for analyzing fecal microbiome data. It effectively addresses challenges related to high-dimensional data and multicollinearity, while providing interpretability [19]. We implemented the ML model based on the training sets and initially confirmed its efficacy in distinguishing IBD from HC and UC from CD. Subsequently, we validate its robustness using separate test sets. This study contributes to the growing body of evidence supporting fecal microbiome analysis for diagnosing and distinguishing IBD [13,14,15,16].

Consistent with previous reports, this study found that CD and UC exhibited lower alpha diversity than that of HC [30,31,32]. Beta diversity analysis revealed relatively distinct differences in phylotype distribution using the Jaccard dissimilarity metric, although some overlap was observed with the thetaYC dissimilarity metric. The Jaccard dissimilarity metric focuses on the presence or absence of taxa across samples, and it does not consider their abundance or relative abundance. In contrast, the thetaYC dissimilarity metric considers both the presence and absence of taxa and their relative abundances. In summary, both patients with CD and UC exhibited distinct bacterial taxa that differentiated them from HC. Previous research has also reported differences in taxa between UC and CD compared to HC, although the extent of these differences varies [30,33].

In patients with IBD, the predominant characteristics included an increase in the Proteobacteria phylum, *Fusobacterium* species, and *Escherichia coli* [31,33,34,35,36], while there was a decrease in protective taxa such as *Faecalibacterium prausnitzii* and *Bifidobacterium* species [32,33,37,38,39,40]. However, information regarding taxonomic differences between IBD and HC varies among the studies conducted thus far, necessitating further clarification regarding the distinctions between UC and CD. Differences among studies, such as sample type, age, sex, dietary habits, disease extent, disease activity, and concomitant therapies, are likely to influence microbial community structure and diversity [30,31,40,41]. Therefore, the application of enumerative information as a diagnostic tool may be limited. This study has clinical value in overcoming these limitations and leveraging the advantages of the sPLS-DA algorithm based on differences in phylotypes to construct an ML model and demonstrate its performance. This study also examined the top 10 genera in HC, IBD, CD, and UC. Notably, we observed differences in the major microbiota between CD and UC, which can provide additional information beyond what was previously reported.

Recent advancements in ML models that leverage fecal microbiome data have shown promising results in IBD diagnosis. For example, using OTUs, the RF algorithm achieved notable performance, with an area under the curve (AUC) of 0.80 and an accuracy of 0.72 for distinguishing IBD from non-IBD groups. Additionally, it attained an AUC of 0.92 and an accuracy of 0.83 for distinguishing between UC and CD [14]. In another study, various feature selection techniques were employed to construct an RF model, which demonstrated acceptable discrimination in external validation, yielding AUCs of 0.74 and 0.76 for diagnosing UC and CD, respectively [18]. Furthermore, a different study developed an RF model using taxonomic profiles at the species level, achieving an AUC of 0.93 and an accuracy of 0.86 for UC diagnosis and an AUC of 0.93 and an accuracy of 0.83 for CD diagnosis [19]. However, the limitations of previous studies include the use of a global data platform, which leads to heterogeneity in disease activity, sample collection, and analysis methods [14,18]. Matching was not conducted to minimize bias in the selection of the non-IBD group [14,18,19]. Additionally, the last study, which employed a multiclass model for various diseases, may have been inappropriate for distinguishing IBD from chronic IBDs [19]. Finally, some studies lacked external validation [14,19]. In our study, the sPLA-DA model exhibited exceptional performance in diagnosing IBD, with a mean accuracy of 0.950. Additionally, it distinguished between UC and CD with a mean accuracy of 0.945, surpassing the performances of previous studies. These advancements in harnessing fecal microbiome data to develop ML models hold great promise for enhancing the diagnosis of IBD diagnosis.

This study had several limitations. First, confounding factors, such as age, sex, diet, and medication, were not fully controlled. Furthermore, when examining the top genera each for IBD, UC, and CD, no clear commonalities were found compared with the significant microbiota increases reported in previous studies [33]. Second, the UC and CD cohorts differed in their characteristics. Patients with UC comprised an inception cohort with fecal samples collected post-diagnosis and pre-treatment, whereas fecal samples of patients with CD were collected at various stages of treatment. To address possible modifiable aspects, we took measures such as discontinuing the use of antibiotics or probiotics before collecting fecal samples. Future studies should consider these factors to enhance our understanding of the microbial diversity in CD and UC. Third, this study lacked external validation, which limited its generalizability. However, during the division of the training and test sets, efforts were made to downsample 100 times with matching age, sex, and sample size. Finally, this study focused on Korean patients diagnosed with IBDs (CD or UC). Microbial communities can vary across geographical regions [42], and the findings of this study may not be directly applicable to other populations.

## 5. Conclusions

In summary, this study successfully developed a prediction model using the sPLS-DA algorithm for diagnosing IBD and differentiating between CD and UC compared with HC, demonstrating good performance. We are optimistic that the ML model developed using fecal microbiome data can contribute to the early diagnoses of CD and UC, facilitating prompt and effective treatments guided by its predictions. However, further external validations across different geographical regions are required to confirm the applicability of the developed model.

## Figures and Tables

**Figure 1 microorganisms-12-00036-f001:**
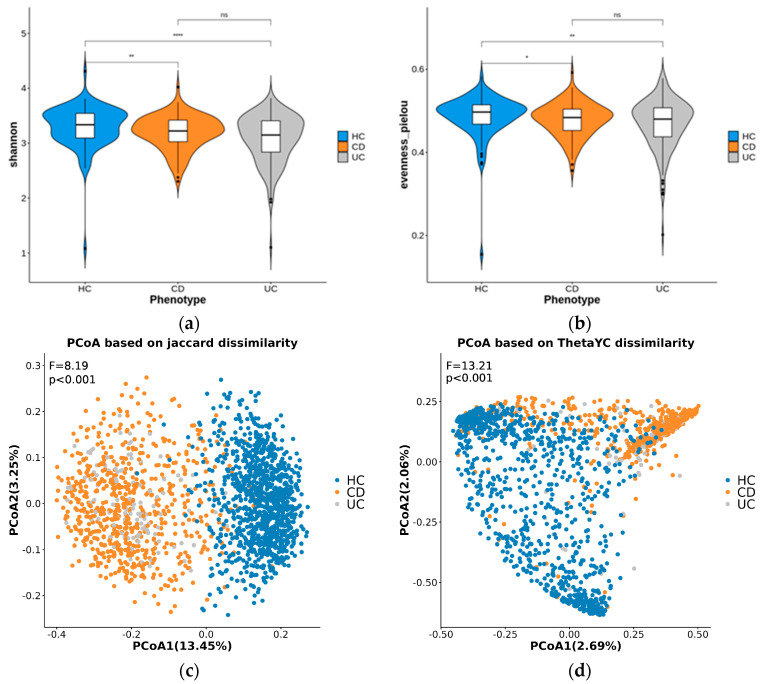
Violin plots displaying α−diversity indices (Shannon (**a**), and Pielou evenness (**b**)) of stool microbiome in the three disease groups. Significance in α−diversity variation between phenotypes was assessed using the Wilcoxon test (*, **, ****, and ns represent *p* < 0.05, *p* < 0.01, *p* < 0.0001, and non−significance respectively). The PCoA plots based on β−diversity indices: Jaccard dissimilarity (**c**) and thetaYC dissimilarity (**d**). F- and *p*-values were calculated by a PERMANOVA test with 1000 permutations.

**Figure 2 microorganisms-12-00036-f002:**
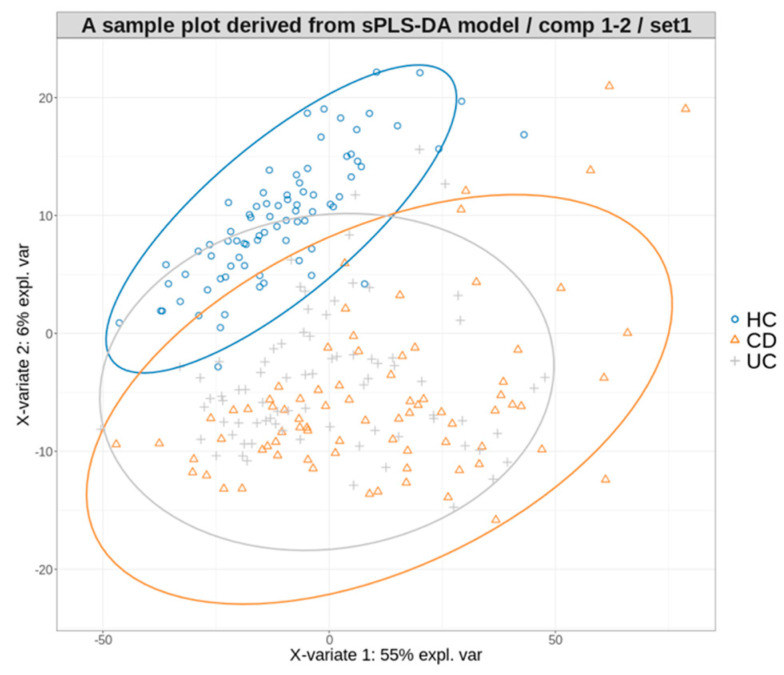
A PLS projection in the subspace defined by the sPLS−DA model’s first two components, developed for multiclass prediction.

**Figure 3 microorganisms-12-00036-f003:**
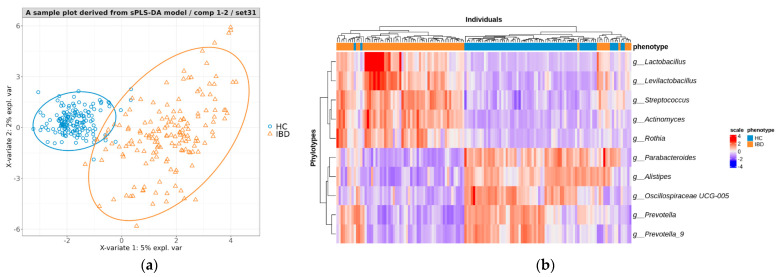
(**a**) A PLS projection in the subspace defined by the sPLS−DA model’s first two components, developed for discriminating between IBD and HC. (**b**) A heatmap representing the abundance of high−contributing phylotype features for predicting the IBD and HC groups in the test set.

**Figure 4 microorganisms-12-00036-f004:**
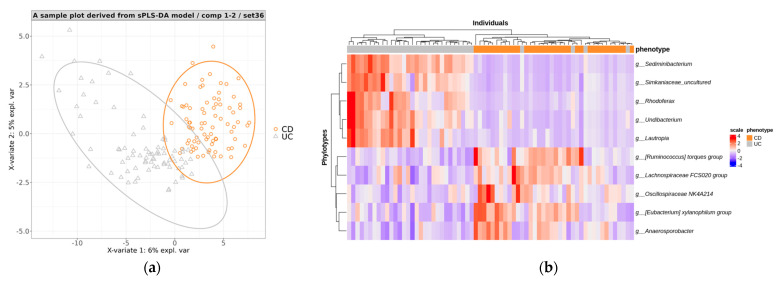
(**a**) A PLS projection in the subspace defined by the sPLS−DA models’ first two components, developed for discriminating between CD and UC. (**b**) A heatmap representing the abundance of high−contributing phylotype features for predicting the CD and UC groups in the test set.

**Table 1 microorganisms-12-00036-t001:** Baseline demographic and clinical characteristics of participants.

	CD (*n* = 671)	UC (*n* = 114)	HC (*n* = 1462)
Age (year), mean ± SD	35.9 ± 13.2	39.9 ± 15.9	45.9 ± 9.2
Male, *n* (%)	483 (71)	84 (73.7)	907 (62)
BMI (kg/m^2^), mean ± SD	22.1 ± 3.8	23.2 ± 3	23.7 ± 3.1
Smoking status, *n* (%)			
Current	93 (13.9)	14 (12.3)	
Former	26 (3.9)	22 (19.3)	
Never	454 (67.7)	77 (67.2)	
Unknown	98 (14.6)	1 (0.0)	
Disease location, *n* (%)	Ileum, 186 (27.7)	Proctitis, 48 (42.1)	
	Colon, 84 (12.5)	Distal, 38 (33.3)	
	Ileocolon, 337 (50.0)	Extensive, 27 (23.7)	
	Ileum + upper GI, 6 (0.9)		
	Colon + upper GI, 1 (0.0)		
	Ileocolon + upper GI, 17 (2.5)		
	Unknown, 40 (6.0)	Unknown, 1 (0.0)	

Values are expressed as *n* (%) unless otherwise specified. SD, standard deviation; BMI, body mass index; GI, gastrointestinal tract.

**Table 2 microorganisms-12-00036-t002:** Information for each processing step.

	OTUs/ Phylotypes	Samples	Total Reads (% of the Raw)
Raw		2255	164,539,577
fastp		2255	157,961,202 (0.96)
HmmUFOtu clustering	88,927	2255	157,865,293 (0.9594)
Chimera removal	83,562	2255	150,585,336 (0.9152)
Taxonomic assignments	67,283	2255	150,549,827 (0.915)
Phylotyping	2525	2255	150,549,827 (0.915)
Abundance table filtering process			
Abundance > 20 k or counts > 10	1526	1853	145,380,866 (0.8836)
Non-bacterial phylotypes	1518	1853	145,353,664 (0.8834)
Domain only	1517	1853	140,691,441 (0.8551)
Abundance > 20 k	1517	1846	140,552,068 (0.8542)

OTU, operational taxonomic unit.

**Table 3 microorganisms-12-00036-t003:** Evaluation metrics from prediction using multiclass models.

	Accuray	CD Sens.	CD Prec.	UC Sens.	UC Prec.	HC Sens.	HC Prec.	AUC
Min.	0.539	0.177	0.292	0.235	0.353	0.677	0.566	0.539
1st Qu.	0.608	0.324	0.443	0.441	0.486	0.934	0.756	0.608
Median	0.637	0.412	0.5	0.544	0.551	0.971	0.823	0.637
Mean	0.638	0.434	0.505	0.53	0.545	0.952	0.814	0.638
3rd Qu.	0.667	0.529	0.559	0.618	0.591	1	0.95	0.667
Max.	0.755	0.824	0.793	0.765	0.735	1	0.944	0.755

Sens, sensitivity; Prec, precision.

**Table 4 microorganisms-12-00036-t004:** Evaluation metrics from prediction using IBD vs. HC models.

	Accuracy	Sensitivity	Specificity	Precision	AUC
Min.	0.89	0.809	0.918	0.91	0.972
1st Qu.	0.941	0.897	0.971	0.97	0.989
Median	0.949	0.919	0.985	0.984	0.993
Mean	0.95	0.918	0.982	0.981	0.992
3rd Qu.	0.963	0.941	1	1	0.996
Max.	0.993	0.985	1	1	1

AUC, area under the curve.

**Table 5 microorganisms-12-00036-t005:** Evaluation metrics from prediction using CD vs. UC models.

	Accuracy	Sensitivity	Specificity	Precision	AUC
Min.	0.853	0.794	0.824	0.846	0.947
1st Qu.	0.927	0.912	0.934	0.93	0.984
Median	0.956	0.941	0.971	0.967	0.991
Mean	0.945	0.941	0.949	0.95	0.988
3rd Qu.	0.971	0.971	0.971	0.971	0.997
Max.	1	1	1	1	1

AUC, area under the curve.

**Table 6 microorganisms-12-00036-t006:** Evaluation metrics calculated in hierarchical manner.

	Accuracy	CD Sens.	CD Prec.	UC Sens.	UC Prec.	HC Sens.	HC Prec.	AUC
Min.	0.873	0.706	0.953	0.912	0.824	0.912	0.773	0.873
1st Qu.	0.922	0.853	0.941	0.924	0.941	0.971	0.85	0.922
Median	0.931	0.882	0.969	0.941	0.97	1	0.895	0.931
Mean	0.936	0.888	0.965	0.933	0.964	0.956	0.891	0.936
3rd Qu.	0.951	0.941	1	0.971	1	1	0.919	0.951
Max.	0.99	1	1	1	1	1	1	0.99

Sens, sensitivity; Prec, precision.

## Data Availability

The sequencing data in fastq format are available upon request.

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
