# Peer review of "A Machine Learning-Based Diagnostic Model for Crohn’s Disease and Ulcerative Colitis Utilizing Fecal Microbiome Analysis"

_microorganisms, 2023, doi:10.3390/microorganisms12010036_

Round 1
Reviewer 1 Report
Comments and Suggestions for Authors
The study uses ML to differentiate between CD, UC, and HC through fecal microbiome analysis. They developed two ML models using sPLS-DA that demonstrated high accuracy and AUC in distinguishing between the groups.
1. Introduction:
A more explicit statement about the limitations of current diagnostic methods for IBD is needed. A more detailed explanation of the inadequacies would strengthen the argument for the necessity of the study.
The mention of diagnostic delay and its consequences is relevant (lines 55-58), but it would be more compelling with data or statistics illustrating the typical delay and its impact on patient outcomes.
The introduction should articulate more clearly the unique contribution of this study. While it suggests that machine learning will be used to create a prediction model, it does not specify how this model will differ or improve upon existing models in the literature.
A table is needed to summarize the previous works by both advantages and limitations.
2. Materials and Methods:
The description of patient recruitment is vague and lacks specificity. The enrollment criteria for participants, the time frame of enrollment, and the demographic characteristics of each cohort should be included to assess the representativeness and generalizability of the study.
It is unclear why UC patients were recruited from a prospective study while CD patients came from a retrospective study.
The choice of the V3-V4 regions of the 16S rRNA gene is not explained.
The parameters for quality control and the thresholds for discarding sequences should be justified.
The rationale behind the chosen thresholds for sample exclusion (lines 127-128) is missing. It is crucial to understand why these particular values were selected to ensure they do not bias the results.
More details on log normalization and batch effect correction are needed (lines 134-137). Specifically, how the batch effects were identified and quantified should be disclosed.
The downsampling approach to address class imbalance (lines 142-143) could introduce bias or loss of valuable information. The manuscript should discuss alternative methods considered and why downsampling was preferred.
The claim of "standard procedures" for feature selection and parameter optimization (line 146) is too generic. The authors should describe these procedures to allow for reproducibility.
3. Results:
The multi-class model showed suboptimal performance for classifying Crohn's Disease (CD) and Ulcerative Colitis (UC), as mentioned in line 206. This could indicate that the features or the model itself may not be capturing the complexities and nuances necessary to differentiate between these two conditions effectively.
In beta diversity analysis, CD and UC samples were not clearly distinguishable (lines 187-188). This lack of distinction could suggest that the microbial profiles are too similar or the dissimilarity indices used do not provide a clear separation.
The hierarchical approach improved classification accuracy, but the method's inherent assumption that the IBD versus HC classification is always correct may not account for misclassifications in the first step, which would propagate errors to the second step.
In the binary classification for distinguishing IBD from HC, twice as many HC samples were included initially (line 220), which may have skewed the training of the model towards a better performance for the HC class.
4. Discussions:
The study acknowledges that it did not fully control for confounding factors such as age, gender, diet, and medication, which can significantly influence the microbiome.
The lack of external validation is a significant limitation. The high accuracy and AUC values are exceptional, but there's a concern that the model may be overfitted to the training data.
Comments on the Quality of English LanguageThere is room for improvement in sentence structure to avoid run-on sentences and enhance readability in the introduction section.
Reviewer 2 Report
Comments and Suggestions for Authors
"A Machine Learning-Based Diagnostic Model for Crohn's Disease and Ulcerative Colitis Utilizing Fecal Microbiome Analysis" by Hyeonwoo Kim et al. is an article concerning the application of a machine learning protocol, complete with data pre-processing, in order to distinguish a pool of subjects affected by IDB (Crohn's disease (CD) and ulcerative colitis (UC)) and healthy controls (HC). Initially, an analysis of the alpha diversity and beta-diversity of the sample pool was performed, showing a clear separation between healthy people (HC subjects) and sick people (IBD subjects). Overall, I agree with the choice of using the sPLS-DA algorithm, which is optimal in selecting the best variables, in this case the bacterial taxa, and in using them to calibrate optimal ML models. However, the analysis has some limitations, starting from using only 16S data and the limited variability in the bioprojects, and, therefore, in the data's origin. Furthermore, the novelty referring to the biological aspect is limited (microbial differences between IBD and HC), as there are already studies in this regard, but it still remains interesting as a test case for the optimization of ML protocols based on sPLS -DA can also be used in other contexts. The overall protocol followed to perform the ML analysis was complete.
Major comment:
1) The three phenotypic groups analyzed present a large gap in initial size (CD: n=671, UC: n=114, HC: n=1, 462), while I'have appreciated the pre-processing of the data performed to homologate and make them comparable (CD: n=670, UC: n=113, HC: n=1,063). However, reducing the final dataset for analysis to (CD: n=113, UC: n=113, HC: n= 113) is too radical and shows too much loss of variability in the data. Given that IBD and HC are optimally divided (in Beta-diversity) while UC and CD overlap to some extent, in creating the binary system, I would ignore the "age" factor and use the data in their entirety (IBD: n=783, HC: n=1,063). Also reducing the HC to 783 to obtain two comparable size pool could be useful, but only through a completely random (unsupervised) selection of HC samples. As a result, the training dataset's robustness should improve significantly.
Minor comment:
1) Consider to use PERMANOVA or ANOSIM statistical analysis in order to confirm the separation in beta-diversity between HC, CD e UC.
Reviewer 3 Report
Comments and Suggestions for Authors
The authors created in this study a clinically helpful machine learning(ML)-based prediction model and attempted to validate its performance in order to differentiate between IBD and healthy controls (HC) and between UC and CD using fecal microbiome analysis. They concluded that this study successfully developed a prediction model using the Sparse Partial Least Squares Discriminant Analysis (sPLS-DA) algorithm for diagnosing IBD and differentiating between CD and UC when compared to HC, demonstrating good performance
My concerns are as follows.
1. Regarding IBD patients, it is described “Patients with CD were recruited from 15 tertiary hospitals in South Korea, participating in a multicenter, retrospective case-control study [20]. Meanwhile, UC patients were sourced from a prospective multicenter study established for UC multiomics research.”. I wonder whether fecal samples were obtained from CD or UC patients with definite diagnoses of each IBD before any pharmaceutical treatment or surgery, which potentially have significant impact on intestinal microbiome. I also wonder whether the inflammation in the bowel were active or controlled at the fecal samplings in IBD patients.
2. More details of CD/UC patients and HC are necessary to be described. It might be better to use age and gender matched subjects among CD, UC, and HC.
3. Although a PLS projection in the subspace defined by the sPLS-DA models's first two components developed for discriminating between IBD and HC, I wonder how it could be possible to discriminate IBD patients and non-IBD patients with various systemic or other gastrointestinal diseases.
4. Although it is clearly demonstrated that the binary prediction models exhibited high accuracy and area under the curve in differentiating IBD from HC, I wonder how this method could be clinically useful to make early diagnoses of CD and UC.
Comments on the Quality of English LanguageNil
Round 2
Reviewer 1 Report
Comments and Suggestions for Authors The author answered all my questions. The manuscript has been sufficiently improved to warrant publication in Microorganisms.Author Response
We would like to express my sincere gratitude for your thoughtful review and constructive feedback on my paper. Your insights have played an extremely crucial role in improving the quality of the manuscript. We deeply appreciate the time and effort you devoted to reviewing our work.
Reviewer 2 Report
Comments and Suggestions for Authors
In its current form, I consider the manuscript acceptable for publication.
Author Response
We would like to express heartfelt gratitude for your thoughtful review and constructive feedback on our paper. Your insights played a crucial role in enhancing the quality of the manuscript, and we deeply value the time and effort you dedicated to reviewing our work.
Reviewer 3 Report
Comments and Suggestions for Authors
Although the authors revised their manuscript based on reviewers’ comments, I still have some concerns as follows.
1. As keywords, “Sparse Partial Least Squares Discriminant Analysis”, ”Crohn disease”, and ”ulcerative colitis” are better to be included instead of ”dysbiosis” and ”16S rRNA sequencing”.
2. The conclusive message in Abstract “This research underscores the diagnostic potential of the ML model based on sPLS-DA utilizing fecal microbiome analysis in distinguishing CD and UC from HC” is not matched with the descriptions in Conclusions in the main text.
3. I wonder why the authors had not revised Abstract in harmony with the significant changes in the main text.
4. The weakest point of this study was that patients in the CD cohort were already diagnosed and undergoing various treatments, who were recruited from retrospective study data. Although the authors described “efforts were made to minimize the effects of modifiable agents by discontinuing antibiotics or probiotics for a minimum of three months before fecal sampling”, I wonder how much this entry criteria were secured. Obviously, an inception cohort comprising newly diagnosed patients should have been used, and fecal samples were collected before the initiation of drug therapy in the CD cohort as was in the UC cohort.
5. Regarding the healthy control group, the descriptions were only “the HC group comprised individuals who had undergone comprehensive physical examinations at the Kangbuk Samsung Hospital Healthcare Screening Center since 2014, showing no signs of gastrointestinal disorders”. Although the authors referred to an article entitled “Association between Neutrophil-to-Lymphocyte Ratio and Gut Microbiota in a Large Population: a Retrospective Cross-Sectional Study”, the characteristics of the HC group are necessary to be described in details. I wonder whether individuals in the group did not have any extra-gastrointestinal disorders, which possibly had a significant influence on gut microbiota.
Comments on the Quality of English LanguageNil
